# Quantitative Characterization of Oxygen-Containing Groups on the Surface of Carbon Materials: XPS and NEXAFS Study

Danil V. Sivkov [1,2,*], Olga V. Petrova [2,*], Sergey V. Nekipelov [2], Alexander S. Vinogradov [1],
Roman N. Skandakov [2], Ksenia A. Bakina [2], Sergey I. Isaenko [3], Anatoly M. Ob'edkov [4], Boris S. Kaverin [4],
Ilya V. Vilkov [4] and Viktor N. Sivkov [2]

[1] Faculty of Physics, Federal State Budgetary Educational Institution of Higher Education "Saint-Petersburg State University", 199034 St. Petersburg, Russia; asvinograd@yahoo.de

[2] Institute of Physics and Mathematics, Komi Science Centre of the Ural Branch of the Russian Academy of Sciences, 167982 Syktyvkar, Russia; nekipelovsv@mail.ru (S.V.N.); scanick@yandex.ru (R.N.S.); tylxen@gmail.com (K.A.B.); sivkovvn@mail.ru (V.N.S.)

[3] Institute of Geology, Komi Science Centre of the Ural Branch of the Russian Academy of Sciences, 167982 Syktyvkar, Russia; isaenko@geo.komisc.ru

[4] G.A. Razuvaev Institute of Organometallic Chemistry of the Russian Academy of Sciences, 603950 Nizhny Novgorod, Russia; amo@iomc.ras.ru (A.M.O.); kaverin@iomc.ras.ru (B.S.K.); mr.vilkof@yandex.ru (I.V.V.)

[*] Correspondence: d.sivkov@spbu.ru (D.V.S.); teiou@mail.ru (O.V.P.); Tel.: +7-812-4284-352 (D.V.S.); +7-821-239-1461 (O.V.P.)

**Abstract:** The results of the comparative quantitative study of oxygen-containing groups adsorbed on the surface of carbonized sponge scaffold (CSS), highly oriented pyrolytic graphite (HOPG), fullerite $C_{60}$ and multi-walled carbon nanotubes (MWCNTs) introduced into a high vacuum from the atmosphere without any pre-treatment of the surface are discussed. The studied materials are first tested by XRD and Raman spectroscopy, and then quantitatively characterized by XPS and NEXAFS. The research results showed the presence of carbon oxides and water-dissociation products on the surfaces of materials. It was shown that main source of oxygen content (~2%) on the surface of HOPG, MWCNTs, and $C_{60}$ powder is water condensed from the atmosphere in the form of an adsorbed water molecule and hydroxyl group. On the CSS surface, oxygen atoms are present in the forms of carbon oxides (4–5%) and adsorbed water molecules and hydroxyl groups (5–6%). The high content of adsorbed water on the CSS surface is due to the strong roughness and high porosity of the surface.

**Keywords:** XPS; NEXAFS; XRD; Raman spectroscopy; HOPG; MWCNTs; fullerite $C_{60}$; carbonized sponge scaffold

## 1. Introduction

Monocarbon compounds such as multi-walled carbon nanotubes (MWCNTs), carbonized biomaterials and fullerite $C_{60}$ are widely used to produce new nanostructured materials. MWCNTs are chemically resistant to aggressive environments, and have large external surface area, exceptional resistance to mechanical stress and thermal conductivity due to their nanostructure, in which carbon atoms are strongly bonded with each other. This structural features make MWCNTs a promising material for a wide range of different applications. Fullerite C60 is a promising compound for creating new materials and nanostructures by hot isostatic pressing (HIP) in an inert medium. One of the popular applications of carbon nanotubes is their use as a substrate for the preparation of catalysts by depositing metal carbides and oxides on their outer surface. In this case, a large specific surface area of nanotubes turns out to be the most demanded parameter. An alternative to nanotubes are natural biomaterials, which have a branched architecture with a large external surface area. During high-temperature treatment in inert atmosphere, such objects undergo graphitization with retaining their volumetric structure.

Modern technologies for the production of fullerites [1–3] and carbon nanotubes [4,5] ensure their production in macroscopic quantities. Extreme biomimetics [6–10] remains to be one of the promising directions in the creation of composites based on thermostable and chemically resistant biopolymers with unique pre-existing three-dimensional (3D) architecture [11,12]. Recently, it was shown [13–16] that the use of natural marine sponge 3D spongin-based frameworks opens the window for special applications also with respect to carbonization. For example, the sponge-scaffold carbonization in argon at temperatures up to 1200 °C leads to the formation of a three-dimensional nanoporous structure of turbostratic graphite, which turned out to be promising for the formation of composite materials for the preparation of catalysts for chemical processes [13]. One of the important applications of MWCNTs and carbonized sponge scaffold (CSS) is the composite material synthesis by depositing metal compounds on their outer graphene-like surface. An important problem in the production of such materials is to determine the mechanism of the coating-layer adhesion to the chemically inert outer surface. This task requires knowledge of the atomic-chemical composition of compounds on the surface. The same problem arises for fullerite $C_{60}$ modified by HIP in an inert argon atmosphere. The conducted studies [17–19] have shown that the presence of oxygen-containing groups on the carbon material surface has a strong influence on the formation of metal-containing coating layers on the MWCNT and CSS outer surface as well as in fullerene $C_{60}$ HIP treatment. In particular, the presence of oxides on the outer surface of carbonized materials can provide good adhesion of metal coatings through the formation of carbon-oxygen-metal bridge bonds during deposition. The main purpose of this work is to determine the quantitative concentration of oxygen atoms and molecular compounds on the surfaces of CSS in comparison with ones on the surface of MWCNTs, highly oriented pyrolytic graphite (HOPG) and $C_{60}$ powder.

## 2. Materials and Methods

### 2.1. Materials

In the study, the following samples were investigated: (i) MWCNTs with average outer diameter of 80 nm and 300 μm length, synthesized in HNL of G.A. Razuvaev Institute of Organometallic Chemistry of RAS according to [17]; (ii) the 99.98% pure fullerite $C_{60}$ with a crystallite size of 0.05–0.5 mm produced by Fullerene-Center, Nizhniy Novgorod, Russia [19]; (iii) HOPG (ZYA grade) produced by SPI supplies, West Chester, PA, USA, and; (iv) the selected spongin scaffolds isolated from H. communis carbonized by the method described in [13]. The sponge was heated at 1200 °C for 1 h in an Ar stream. After the heat treatment the sponge volume was reduced by a factor of three, while its fibrous three-dimensional structure was retained. The compressive strength of the carbonized sponge was 1.3 MPa at a 0.1119 g/cm$^3$ density and the Brunauer–Emmett–Teller (BET) specific surface area increased by more than two orders of magnitude from 3.45 m$^2$/g to 425 m$^2$/g due to the mesoporous surface development [13].

### 2.2. Characterization

The samples were studied by set of complementary methods: X-ray diffractometry (XRD), Raman, near edge X-ray absorption fine structure (NEXAFS) [20] and photoelectron (XPS) spectroscopy [21,22].

The XRD analysis of materials was performed with $CuK_{\alpha}$-radiation using a Bruker D8 Discover X-ray diffractometer in the $\theta$–$2\theta$ geometry with a Gobel mirror, an equatorial Soller slit with the angular divergence of 2.5°, and an exit slit of 1.5 mm. Scanning was performed in 0.02° increments at 2–2.5°/min scanning speed. The resulting diffractograms were processed with the EVA software using the PDF-2 (2012)—a database of diffraction data.

Raman spectroscopy was conducted at room temperature in the 80–3500 cm$^{-1}$ interval using a Horiba-Yvon Jobin LabRam HR800 spectrometer equipped with 600 g/mm grating and Ar laser with a 1 mW and 488 nm wavelength. Lenses of 50× and 100× were used for analysis. The spectrometer was fitted with neutral filters to limit the laser radiation

power. The spectral and spatial resolution was about 1 cm$^{-1}$ and 1 μm, respectively. Sample damage was an issue in fullerite C$_{60}$ Raman measurements, so the power on the C60 sample was well below 1 mW, and its Raman spectra were measured at the minimum recording time. For HOPG, MWCNTs and CSS, registration times of up to 60 s were used with no visible damage or changes in signal shape. Raman spectra of all samples were performed at different points for each sample.

The electronic structure of all samples was characterized by the NEXAFS spectroscopy. These studies were performed using the synchrotron radiation of the Russian-German soft X-ray beamline at BESSY II synchrotron radiation facility (Berlin, Germany). In all NEXAFS spectra measurements a total electron yield (TEY) mode was used. The energy positioning of C 1s NEXAFS spectra elements were performed using the narrow π*-resonance at 285.38 eV in the C 1s spectrum of HOPG [23]. The spectral dependence of photon flux was determined as a result of the division of TEY signal from clean Au photocathode and Au atomic X-ray absorption cross section [24] in accordance with [17,25]. The photon energy resolution was better than 0.05 eV. The ex situ preporation of the samples of MWCNTs, C$_{60}$ and CSS for X-ray absorption measurements was performed by pressing the powder of the test material into the clean surface of an indium or copper plate in air. The sample in the form of a 2 × 2 mm plate was attached with silver glue to the surface of the copper holder. The X-ray incidence angles were chosen between 40–50 degrees to the holder surface.

The X-ray photoelectron studies were performed at the resource Centre "Physical methods of surface investigation" of the Science Park of St. Petersburg State University (St. Petersburg, Russia). XPS analysis was performed using a Thermo Scientific ESCALAB 250Xi electron spectrometer equipped with an electron-ion charge compensation system to neutralize the sample charging during experiments. The X-ray tube with AlKα radiation (1486.6 eV) was used as a source of ionizing radiation. The survey and in the are of internal shells photoelectron (PE) spectra were obtained at pass energy of 100 eV and 50 eV, respectively. The ESCALAB 250 Xi spectrometer software was used to process the experimental data.

## 3. Results and Discussion

The results of preliminary tests of the samples of HOPG, MWCNTs, fullerene C$_{60}$ and CSS by Raman method are shown on Figure 1. The MWCNT Raman spectrum contains the main peak G (1586 cm$^{-1}$), and in contrast to the HOPG spectrum—the peak D (1361 cm$^{-1}$), characteristic of noncrystalline materials. The ratio of the intensities of D and G peaks (I(D)/I(G)) demonstrates the degree of sample graphitization. The lower the ratio, the higher the graphitization and the fewer defects and impurities are in the graphene layers of MWCNTs. For the investigated nanotube samples, this value is 0.45 and corresponds to high-purity MWCNT [26].

The CSS Raman spectrum includes the D, G, and 2D bands. The analysis of the energy position and intensities of these bands, as well as the I(D)/I(G) ratio indicates increasing clustering and the presence of sp$^2$ chains in CSS, according to model proposed by Ferrari and Robertson [27]. More precisely, it can be said that the material evolves from amorphous carbon toward nanocrystalline graphite with mixed bond based on sp$^2$ and sp$^3$ hybridization. The average graphite nanocrystallite size is ~3 nm [13]. Two small extra peaks A$_1$ (~450 cm$^{-1}$) and A$_2$ (850 cm$^{-1}$) in CSS Raman spectrum are typical of low sp$^3$ amorphous graphite [28,29]. The spectrum of fullerite shows all the peaks characteristics of pure C$_{60}$ [30,31].

The diffractograms of MWCNTs, fullerite C$_{60}$ and CSS are shown on Figure 2. Fullerite XRD patterns consist of a sequence of narrow peaks, whose positions correspond to a face-centered molecular crystal with a lattice constant of 1.417 nm and a distance between C$_{60}$ molecules of 1.002 nm [30,32,33].

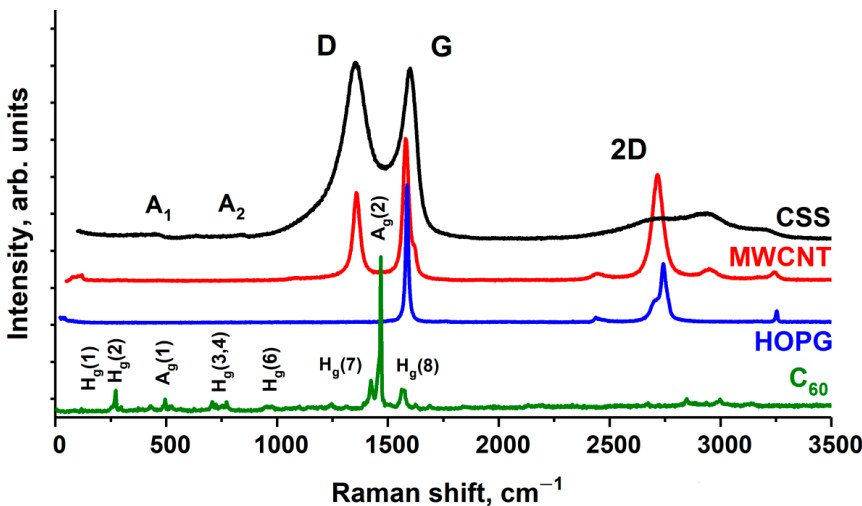

**Figure 1.** The Raman spectra of the fullerite $C_{60}$, HOPG, MWCNTs and CSS.

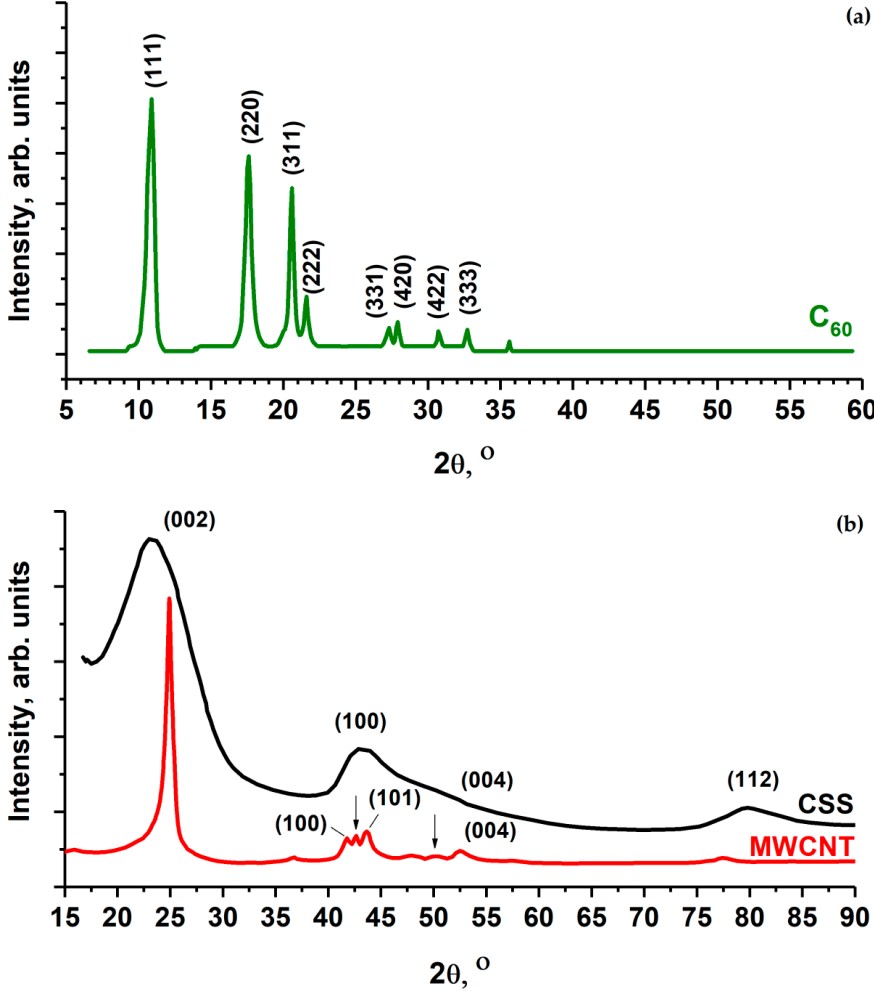

**Figure 2.** XRD characterization of the $C_{60}$ (**a**) and MWCNT, CSS powders (**b**). The arrows indicate the positions of diffraction peaks of γ-Fe from the iron catalyst.

The XRD pattern of MWCNT (Figure 2b) consists of four peaks characteristic of graphite: (100), (002), (004) and (112). Moreover, diffraction peaks marked with an arrow in Figure 2b prove the presence of residual iron catalyst (γ-Fe) in the MWCNT inner volume.

The XRD patterns of the CSS sample (Figure 2b) contain broad peaks, and only the strong (002), (100) and (112) peaks may be reliably distinguished. The peaks (100) and (112) are strongly asymmetric. The XRD patterns of turbostructurally disordered carbon contain peaks with similar shapes [13].

The XRD and Raman spectroscopy methods allow for obtaining the bulk structural characteristic of the samples as a whole. NEXAFS [20] and XPS [21,22], in contrast, characterize the changes in the system of electron states at the level of an individual molecular cluster on the surface of the samples. The high sensitivity of the C 1s fine structure absorption spectra of carbon-containing materials to the presence of C–O, C–O–C, C=O linear and carbonate $CO_3$ planar functional groups is due to the high oscillator strength of the C $1s \rightarrow \pi^*$ transitions in these atomic groups. Therefore, even at low concentrations of carbon oxides, their presence is observed in the C 1s NEXAFS spectra of the samples.

The C 1s NEXAFS spectra of HOPG, MWCNTs, fullerene $C_{60}$ and CSS are shown on Figure 3, where the vertical dashed lines point out the positions of the peaks corresponding to C $1s \rightarrow \pi^*$ transitions in the atomic groups C=O (288.4 eV), C–O–C (287.2 eV), C–O (286.4 eV) [25,34,35] and in anion $[CO_3]^{2-}$ [36]. In the C 1s absorption spectrum of fullerite and pyrolytic graphite, there is no fine structure due to the presence of the carbon oxides mentioned above. Consequently, water molecules are physically adsorbed on the surface of $C_{60}$ powder and HOPG without the formation of a chemical bond with carbon. This is confirmed by the data from [33,37–39], where it was shown that at temperatures below 470 K physically adsorbed water molecules remain on the fullerite surface, and at temperatures above 570 K carbon oxides are formed [32].

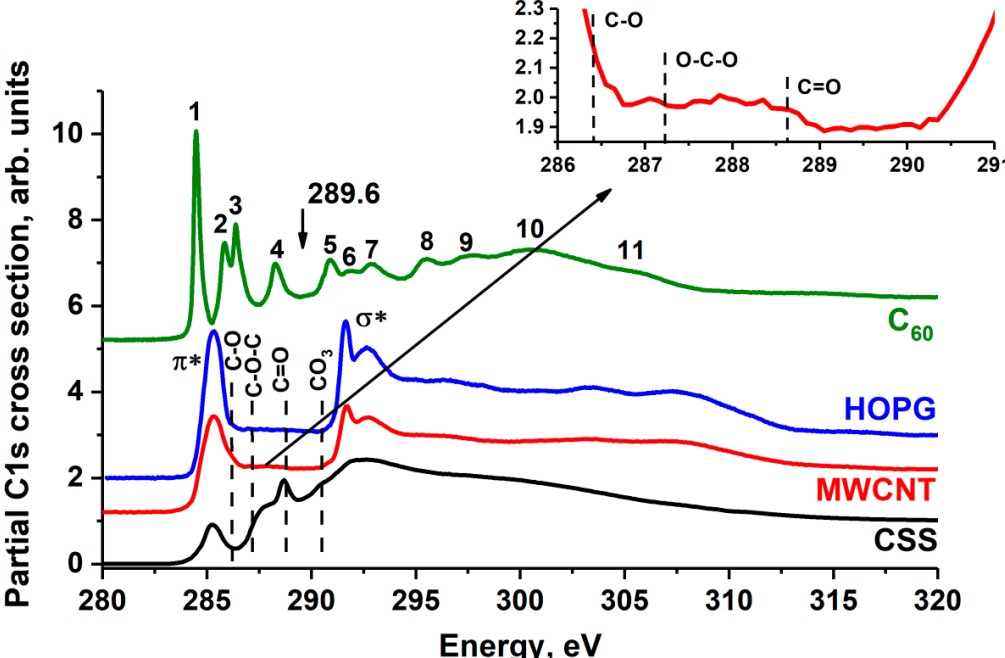

**Figure 3.** C 1s NEXAFS spectrum of HOPG, MWCNT, CSS and fullerite $C_{60}$. The HOPG spectrum was measured at 45° incidence angle to the sample surface. The spectra are normalized to the same level of the C 1s continuous absorption at the 320 eV photon energy. The arrow indicates the position of the fullerite $C_{60}$ C 1s absorption edge (289.60 eV) [40]. In the insert, the C 1s NEXAFS spectra of MWCNT in energy region 286–291 eV is shown.

The fine structure C 1s absorption spectrum of the MWCNTs is in good agreement with the data from elsewhere [41–43]. In addition to the $\pi^*$ and $\sigma^*$ absorption peaks characteristic of the HOPG spectrum, the MWCNT C 1s NEXAFS spectrum exhibits additional structures in the 287–290 eV photon energy interval. In the insert in Figure 3, additional peaks associated with the transitions of the C 1s core electrons to unoccupied $\pi^*$ molecular orbital

of the C–O, C–O–C and C=O functional groups are indicated by dashed lines. Since the content of carbon atoms of these groups in MWCNTs is very small, their contribution to the C 1s NEXAFS spectrum of MWCNTs appears as the low intensity band. According to their XPS data (Figure 4), the contribution of these oxygen groups is not more than 2 at.%. The C 1s NEXAFS spectra of CSS exhibit the broad band at ~287.2 eV (C–O, C–O–C), peak at 288.5 eV (C=O) and small peak at 290.3 eV, which corresponds to the anion $[CO_3]^{2-}$. Thus, the NEXAFS spectroscopy data demonstrate the presence of carbon oxides on the CSS surface. According to its XPS data (Figure 4), the oxides contribution is not more than 10%.

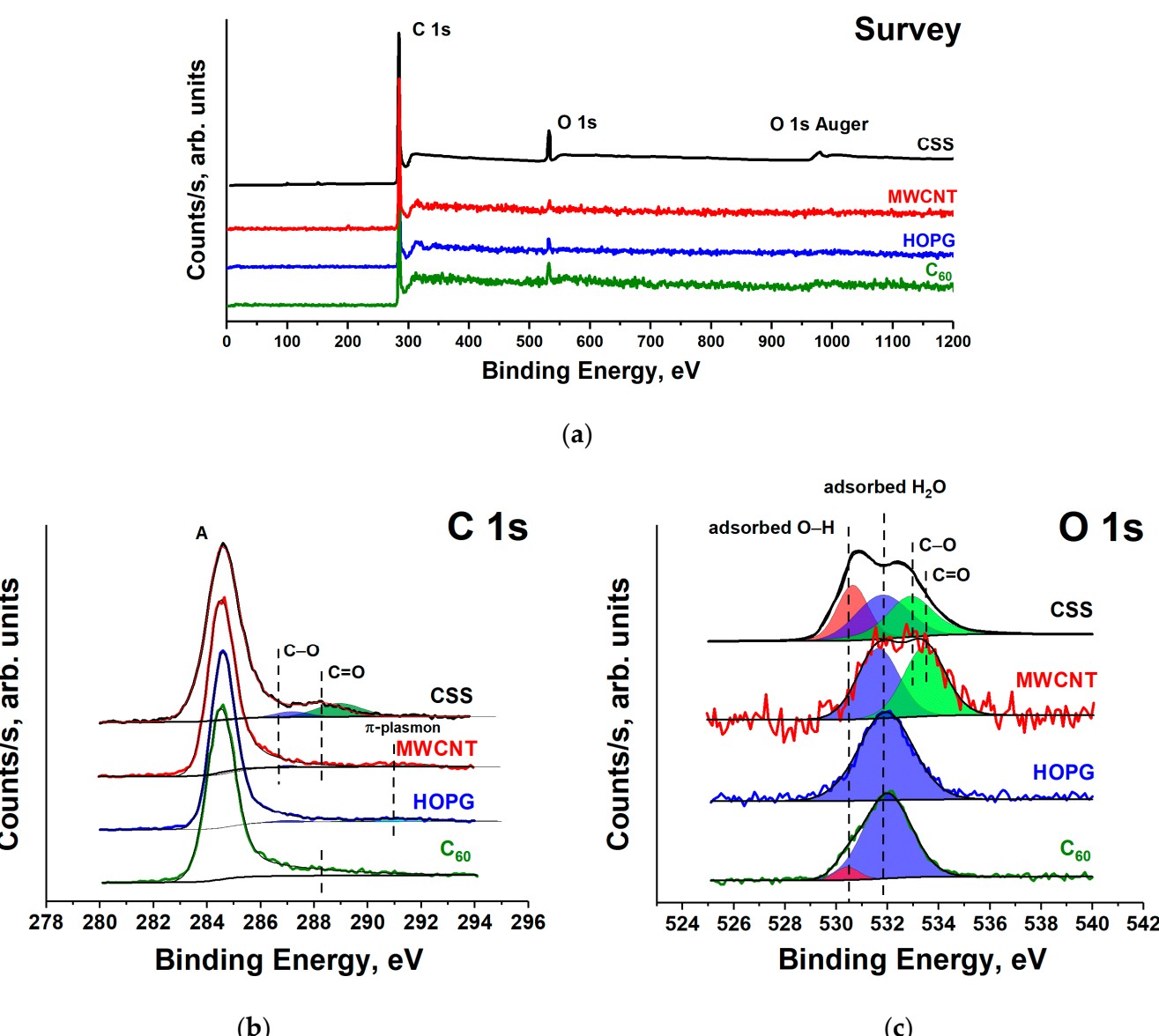

**Figure 4.** The survey (**a**), C1s (**b**) and O1s (**c**) XPS spectra of HOPG, MWCNT, fullerene $C_{60}$ and CSS.

Figure 4a shows the survey XPS spectra of the HOPG, MWCNT, fullerene $C_{60}$ and CSS subjected to long-term atmospheric exposure. The survey spectra contain only two peaks: an intense C 1s and a weak O 1s. The availability of the O 1s peak in the survey XPS spectra indicates the presence of oxygen compounds on the sample surface. According to the XPS measurements, the oxygen relative atomic concentration on the HOPG, MWCNTs and $C_{60}$ powder surfaces is equal to 3–4%, and on the CSS surface—to 10%. Taking into account the absence of a fine structure associated with carbon oxides in the C 1s NEXAFS spectra of fullerite $C_{60}$ and HOPG (Figure 3), it can be assumed that the O 1s peaks in

their XPS spectra are associated with adsorbed $O_2$ and $H_2O$ molecules on the surface. In all XPS C 1s spectra (Figure 4b), one intense asymmetric peak is observed, and only in the HOPG spectrum in the region of high binding energies (BEs) a $\pi$-plasmon satellite is detected [44]. In the spectra of the studied compounds, there is no structure associated with carbon oxides.

Figure 4 shows O 1s XPS spectra of the studied compounds. All spectra contain an intense peak with BE of ~532 eV. In the fullerite $C_{60}$ spectrum, it is the additional low-intensity peak with energy of 530.6 eV, and in the MWCNT spectrum—the band with energies of 533.5 eV. In the 1s XPS CSS spectrum, three peaks of approximately equal intensity at BE of 530.6 eV, 531.8 eV and 533.0 eV can be distinguished (Figure 4b). According to the [32,40,41] O 1s BE of $H_2O$ molecules adsorbed on the fullerite surface is in the interval of 532–34 eV, which is much lower than BE of a free $H_2O$ molecule (535.75 eV) [45]. The BE of the O 1s electrons of $H_2O$ molecule adsorbed on the surface of a solid state generally shifts to the lower BE region, compared to a free molecule. This shift is due to the surface shielding effects, since the inner shell of $H_2O$ is more effectively shielded by surface electrons than by the electrons of $H_2O$ molecule itself. In the XPS spectra of a water molecule adsorbed on a metal surface, two O 1s peaks at BEs of ~532.5 eV ($H_2O$) and ~531 eV (hydroxyl groups) are usually observed [46]. The presence of bound hydroxyl groups enhances water binding [45], as the $H_2O$–OH H-bonding is stronger than the $H_2O$–$H_2O$ H-bonding. Thus, two O 1s peaks with BE of 530.8 eV and 532.4 eV were observed for Cu (110) [45] and at 530.8–532.3 eV for Ru [47]. Moreover, the formation of a $H_2O$ bilayer was observed on the surface of metal [48], graphene [49] and fullerite $C_{60}$ [40] under vacuum of $10^{-9}$ torr. Thus, a comparison of our XPS data with the results of other studies suggests that the low-energy component (530.6 eV) of the O 1s XPS spectrum of $C_{60}$ is associated with hydroxyl groups, and the 532.1 eV line is associated with $H_2O$ from the surface layer of water. The HOPG O 1s XPS spectrum contains one peak at 532.0 eV, which can be also associated with adsorbed water.

The C 1s photoelectron spectrum of MWCNTs (Figure 4b) is composed of four bands: the high intensity peak A at photon energy of 284.5 eV (corresponding to BE of carbon atoms in graphite), the two bands—286.4 eV and 287.6 eV (corresponding to BE of carbon atoms singly and doubly bonded to oxygen atoms, respectively) and 290.8 eV band—a $\pi$-plasmon satellite of the C 1s line in aromatic compounds. The O 1s XPS spectrum of MWCNTs was fitted by two peaks (Figure 4c) with O1s BEs of 531.7 eV and 533.5 eV, assigned to of the oxygen atom in the C=O and C–O functional groups, respectively [50]. However, taking into account the previous analysis of the $C_{60}$ and HOPG spectra, the peak at BE of 531.7 eV in the O 1s spectrum of MWCNTs should be associated with adsorbed water molecules.

The situation changes for the C 1s and O 1s XPS spectra of the CSS, as shown in Figure 4b,c. The C 1s XPS spectrum of the CSS is composed of three bands. The band A at BE of 284.5 eV corresponds to carbon atoms in turbostratic graphite, the band at BE of 286.4 eV is connected with the presence of carbon atoms singly bonded to oxygen atoms (C–O) in MWCNT, and the band with BE of 287.6 eV represents the signal from carbon atoms doubly bonded to oxygen atoms (C=O). This identification is according to C 1s NEXAFS data. The O 1s spectrum is composed of three bands at BEs of 530.6 eV, 531.8 eV and 533.0 eV. The O 1s BEs of many compounds fall within a very narrow range, resulting in overlap of a large number of certain spectral components. The CSS C 1s NEXAFS spectrum contains the high energy peak (~532 eV) associated with C–O, C–O–C, C=O bonds and anion $[CO_3]^2$. Moreover, oxygen will always be present at surfaces exposed to the atmosphere due to water adsorption. Therefore, the low-energy peaks in the O 1s spectrum of CSS, as in the case of the fullerite $C_{60}$, are associated with hydroxyl groups (530.6 eV) and with $H_2O$ from the surface water layer (531.8 eV). The water presence on the CSS surface (~5–6 at.%) compared to $C_{60}$, HOPG and MWCNT (~2 at.%) is probably due to high roughness and specific area of its surface due to the formation of abundant nanopores [13].

## 4. Conclusions

In this work the quantitative concentration of oxygen atoms and carborne oxides compounds on the surfaces of HOPG, MWCNT, CSS and $C_{60}$ powders was determined. It was shown that the main source of oxygen content in the amount of approximately 2 at.% on the surface of HOPG, MWCNT, and $C_{60}$ powder is water condensed from the atmosphere in the form of an adsorbed water molecule and hydroxyl group. At the same time, a small amount of oxygen is contained in the form of carbon oxides on the MWCNT surface. The situation is fundamentally different for the CSS surface, where the concentrations of oxygen atoms as a part of carbon oxides, and in the form of adsorbed water molecules and hydroxyl groups are close and equal to 4–5% and 5–6%, respectively. The high content of adsorbed water on the CSS surface is maybe due to the strong roughness and high porosity of the surface, which, according to the BET method, was $425 \pm 30$ m$^2$ [13]. The obtained data are important for the development of methods for the modification of carbon materials and the synthesis of nanostructured composites based on them.

**Author Contributions:** Conceptualization, D.V.S. and O.V.P.; research supervision, D.V.S., O.V.P., S.V.N. and V.N.S.; writing—original draft, D.V.S., O.V.P., S.V.N., A.S.V. and V.N.S.; XRD measurement and their data analysis, I.V.V., B.S.K. and A.M.O.; Raman spectroscopy measurements, S.I.I.; XPS and NEXAFS spectroscopy measurements and its data interpretation, D.V.S., S.V.N., O.V.P., A.S.V., K.A.B., R.N.S. and V.N.S. All the authors were actively involved in discussing the results, writing and drafting the manuscript. All authors have read and agreed to the published version of the manuscript.

**Funding:** The work was supported by the Grant of the President of the Russian Federation MK-3796.2021.1.2; by the Ministry of Science and Higher Education of Russia under Agreement N 075-15-2021-1351 in part of research on NEXAFS spectroscopy. The reported study was funded by RFBR, project number 19-32-60018, and the Komi Republic, project number 20-42-110002 r-a; the state research target for the G.A. Razuvaev Institute of Organometallic Chemistry of RAS, theme No. 45.4; the bilateral program of the RGBL at BESSY II.

**Institutional Review Board Statement:** Not applicable.

**Acknowledgments:** Analytical research was undertaken using equipment of NRC «Kurchatov Institute»—IREA Shared Knowledge Center, the Center for Studies in Surface Science of Research Park of St. Petersburg State University. The work was carried out using the equipment of the center for collective use "Analytical Center of the IOMC RAS" with the financial support of the grant "Ensuring the development of the material and technical infrastructure of the centers for collective use of scientific equipment" (RF—2296.61321X0017, No. 075-15-2021-670).

**Conflicts of Interest:** The authors declare no conflict of interest. The funders had no role in the design of the study; in the collection, analyses, or interpretation of data; in the writing of the manuscript, or in the decision to publish the results.

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
