# Peer review of "Quantitative Characterization of Oxygen-Containing Groups on the Surface of Carbon Materials: XPS and NEXAFS Study"

_applsci, doi:10.3390/app12157744_

Round 1

Reviewer 1 Report

This paper deals with quantitative characterization of oxygen-containing groups on the surface of carbon materials.

Very clean spectra were measured for multiple measurement methods, and carefully described. I considered worthy of publication. 

The following should be modified;

1) They said " The main purpose of this work is to determine the quantitative concentration of oxygen atoms and molecular compounds on the surfaces of CSS in comparison with ones on the surface of MWCNTs and C60 powder and highly oriented pyrolytic graphite (HOPG).". But, very scant description of the CSS. They just wrote "the selected spongin scaffolds isolated from H. communis carbonized by the method described in [18]". I cannot image the CSS in this manuscript. The author should describe the CSS in such a way that the reader can understand such important material without having to read the cited references.

And by the way, I don't understand why the order of the descriptions about CSS is third.

2) What is the average length and thickness of the MWCNT?

And how is it fixed during measurement?

Are the tubes oriented in alignment or in pieces?

3) Incident angle of SR in the NEXAFS measurement was not described.

The spectral profile of HOPG (and perhaps MWCNT) is varied on the incident angle. Therefore, the information on the incident angle is essential.    

Author Response

Point 1: They said " The main purpose of this work is to determine the quantitative concentration of oxygen atoms and molecular compounds on the surfaces of CSS in comparison with ones on the surface of MWCNTs and C60 powder and highly oriented pyrolytic graphite (HOPG).". But, very scant description of the CSS. They just wrote "the selected spongin scaffolds isolated from H. communis carbonized by the method described in [18]". I cannot image the CSS in this manuscript. The author should describe the CSS in such a way that the reader can understand such important material without having to read the cited references.

And by the way, I don't understand why the order of the descriptions about CSS is third.

Response 1: An inaccuracy in the reference was corrected from [18] to [13] and a description of CSS was added to the text of the article. "The sponge was heated at 1200 °C for 1 hour in an Ar stream. After the heat treatment the sponge volume was reduced by a factor of three, while its fibrous three-dimensional structure was retained. The compressive strength of the carbonized sponge was 1.3 MPa at a density of 0.1119 g/cm3 and the Brunauer-Emmett-Teller (BET) specific surface area increased by more than two orders of magnitude from 3.45 m2 /g to 425 m2/g due to the development of mesoporous surface [13]."

The fact that the CSS description is third in the list is not particularly meaningful. It is simply a list of the researched materials.

Point 2: What is the average length and thickness of the MWCNT?

And how is it fixed during measurement?

Are the tubes oriented in alignment or in pieces?

Response 2: The parameters of the MWCNTs is added in the section 2.1.
"with average outer diameter of 80 nm and 300 μm length"

The following information is added in the section 2.2. "The samples of MWCNTs, C60 and CSS for X-ray absorption measurements were prepared ex situ in air by pressing the powder of the test material into the clean surface of a copper or indium plate. The sample in the form of a 2x2 mm plate was attached with silver glue to the surface of the copper holder. The X-ray incidence angles were chosen between 40-50 degrees to the holder surface." Therefore, the structure of the nanotube samples, as well as C60 and CSS ones, were completely disoriented.

Point 3: Incident angle of SR in the NEXAFS measurement was not described.

The spectral profile of HOPG (and perhaps MWCNT) is varied on the incident angle. Therefore, the information on the incident angle is essential.

Response 3: The authors agree with the comment. In the Figure 3 caption the text "The HOPG spectrum was measured at 45° incidence angle to the sample surface" is added.

Reviewer 2 Report

The authors present significant data about quantitative characterization of oxygen-containing groups on 2 the surface of carbon materials: XPS and NEXAFS study. A very interesting data comparing different materials containing oxygen was conducted, the carbonized sponge scaffold (CSS), highly oriented pyrolytic graphite (HOPG), 19 fullerite C60 and multi-walled carbon nanotubes (MWCNTs). Some questions emerged during the reading of manuscript and results presented by the authors.

Find below:

1) The authors need justify better in the introduction section why carbonized materials are chosen principally based on specific surface area and 3D structures.

2) Why are the influences of oxygen content on the carbonaceous materials on applications presented actually for selected materials elected in this study. This point needs to be highlighted in the introduction, objective, and conclusion of this work.

3) Please verify the description of techniques employed more details need to be given, on XRD as an example the scan step, speed, and font employed need to be informed. The same for Raman spectroscopy, number of scans per measurement.  

4) About the choice of XPS and NEXAFS, will be very interesting for the authors to comment on the limitation of conventional techniques to characterize in situ of oxygen content on this type of material and also relate this content to desired applications.

5) It Will be very interesting also compare these advanced techniques with SEM/EDS mapping (highlighting the oxygen content).

6) The water content could be quantified by TG analysis. The pre-treatment of these samples needs to be detailed in the experimental section. The same starting conditions were employed for all samples?

7) The authors conclude your manuscript and the main differences in characteristics such as rugosity and porosity, characteristics given by textural analysis, however, without experimental results, it's just assumptions. Please give the N2 adsorption/desorption isotherms and discuss better this point.

Author Response

Point 1: The authors need justify better in the introduction section why carbonized materials are chosen principally based on specific surface area and 3D structures.

Response 1: Appropriate text is added to the Introduction. "Fullerite C60 is a promising compound for creating new materials and nanostructures by hot isostatic pressing (HIP) in an inert medium. One of the popular applications of carbon nanotubes is their use as a substrate for the preparation of catalysts by depositing metal carbides and oxides on their outer surface. In this case, a large specific surface area of nanotubes turns out to be the most demanded parameter. An alternative to nanotubes are natural biomaterials, which have a branched architecture with a large external surface area. During high-temperature treatment in inert atmosphere such objects undergo graphitization with retaining their volumetric structure."

Point 2: Why are the influences of oxygen content on the carbonaceous materials on applications presented actually for selected materials elected in this study. This point needs to be highlighted in the introduction, objective, and conclusion of this work.

Response 2: Appropriate text is added to the Introduction. "In particular, the presence of oxides on the outer surface of carbonized materials can provide good adhesion of metal coatings through the formation of carbon-oxygen-metal bridge bonds during deposition."

Point 3: Please verify the description of techniques employed more details need to be given, on XRD as an example the scan step, speed, and font employed need to be informed. The same for Raman spectroscopy, number of scans per measurement.

Response 3: Relevant information has been added to the Chapter 2.2. "Scanning was performed in 0.02° increments at a scanning speed of 2-2.5°/min." and "Sample damage was an issue in fullerite C60 Raman measurements, so the power on the C60 sample was well below 1 mW, and its Raman spectra were measured at the minimum recording time. For HOPG, MWCNTs and CSS registration times of up to 60 sec were used with no visible damage or changes in signal shape. Raman spectra of all samples were performed at different points for each sample."

Point 4: About the choice of XPS and NEXAFS, will be very interesting for the authors to comment on the limitation of conventional techniques to characterize in situ of oxygen content on this type of material and also relate this content to desired applications.

Response 4: Relevant information can be found in section 3. "The XRD and Raman spectroscopy methods allow to obtain the bulk structural characteristic of the samples as a whole. NEXAFS [20] and XPS [21,22], on the other hand, characterize the changes in the system of electron states at the level of an individual molecular cluster on the surface of the samples. The high sensitivity of the C 1s fine structure absorption spectra of carbon-containing materials to the presence of C–O, C–O–C, C=O linear and carbonate CO3 planar functional groups is due to the high oscillator strength of the C 1s → π* transitions in these atomic groups. Therefore, even at low concentrations of carbon oxides, their presence is observed in the C 1s NEXAFS spectra of the samples."

Point 5: It Will be very interesting also compare these advanced techniques with SEM/EDS mapping (highlighting the oxygen content).

Response 5: It is not possible to determine the presence of water molecules on the surface of a powdered material by the SEM method. EDS, on the other hand, although characterised by high oxygen sensitivity, is a volumetric method and can only provide quantitative information on the content of oxygen atoms, but not water molecules.

Point 6: The water content could be quantified by TG analysis. The pre-treatment of these samples needs to be detailed in the experimental section. The same starting conditions were employed for all samples?

Response 6: Unfortunately, the TG analysis is not suitable for HOPG, MWCNT and fullerite samples, as: (1) it is performed in the inert (Ar or N2) flow, and not in a high vacuum; (2) according to photoelectron spectroscopy data the water molecule content on the outer surface of the samples is only 1-2 atomic % in relation to 1-2 graphene layers. At a gram weighting of the investigated sample in which the number of graphene layers will be more than 2-3 orders of magnitude, the weight content of water molecules on the surface will be not more than 0.01 mg, i.e. at the limit of mass loss measurement error. The measurements for CSS are planned and the results will be discussed together with the TG data of the original sponge in our next paper.

Point 7: The authors conclude your manuscript and the main differences in characteristics such as rugosity and porosity, characteristics given by textural analysis, however, without experimental results, it's just assumptions. Please give the N2 adsorption/desorption isotherms and discuss better this point.

Response 7: The authors agree with the reviewer's comment that the conclusion should not be of an absolute nature and have made an appropriate change in the text, and have given data on porosity measured by the Brunauer-Emmett-Teller (BET) method for the N2 adsorption/desorption "high porosity of the surface, which, according to the BET method, was 425 ± 30 m2 [13]. " A discussion of the N2 adsorption/desorption isotherms was not planned within this article, but will be carried out in the next article dealing with the carbonisation processes of the original sponge.